# Unilateral Lung Agenesis: A Systematic Review of Prevalence, Anatomical Variants, and Clinical Implications

**DOI:** 10.3390/diagnostics15172272

**Published:** 2025-09-08

**Authors:** Mathias Orellana-Donoso, Mariano Barrenechea-Salvador, Joaquín Caro-Navarro, Matías Cervela-Díaz, Cristian Chacón-Ortiz, Nicolás Claudet-Córdoba, Juan Sanchis-Gimeno, Pablo Nova-Baeza, Juan José Valenzuela-Fuenzalida, Alejandra Suazo-Santibañez, Iván Valdes-Orrego, Gloria Cifuentes-Suazo, Jose E. Leon-Rojas

**Affiliations:** 1Escuela de Medicina, Universidad Finis Terrae, Santiago 7501015, Chile; mathias.orellana@unab.cl (M.O.-D.); mbarrenecheasalvador@uft.edu (M.B.-S.); jcaron@uft.edu (J.C.-N.); mcervelad@uft.edu (M.C.-D.); cchacono@uft.edu (C.C.-O.); nclaudetc@uft.edu (N.C.-C.); 2Facultad de Medicina y Ciencia, Universidad San Sebastián, Lota 2465, Santiago 7510157, Chile; 3Giaval Research Group, Faculty of Medicine, University of Valencia, 46010 Valencia, Spain; juan.sanchis@uv.es; 4Departamento de Morfología, Facultad de Medicina, Universidad Andrés Bello, Santiago 8320000, Chile; pablo.nova@usach.cl (P.N.-B.); juan.kine.2015@gmail.com (J.J.V.-F.); 5Departamento de Ciencias Química y Biológicas, Facultad de Ciencias de la Salud, Universidad Bernardo O’Higgins, Santiago 8370993, Chile; 6Department of Morphology and Function, Faculty of Health Sciences, Universidad de las Américas, Santiago 8370040, Chile; alej.suazo@gmail.com; 7Facultad de Ciencias de la Salud, Universidad Autónoma de Chile, Santiago 8910060, Chile; ivan.valdes.orrego@gmail.com; 8Facultad de Medicina, Carrera de Odontología, Universidad Católica de la Santísima Concepción, Av. Alonso de Ribera 2850, Concepción 4090541, Chile; gbcifuentess@gmail.com; 9Cerebro, Emoción y Conducta (CEC) Research Group, Escuela de Medicina, Universidad de las Americas (UDLA), Quito 170513, Ecuador

**Keywords:** unilateral lung agenesis, unilateral lung aplasia, unilateral lung absence

## Abstract

**Background:** Unilateral lung agenesis (ULA) is a rare congenital anomaly characterized by the complete absence of one lung, often accompanied by cardiovascular, skeletal, or gastrointestinal malformations. Despite its clinical significance, evidence of prevalence, anatomical variants, and outcomes remain fragmented. This systematic review aimed to synthesize existing data on ULA’s prevalence, anatomical classifications, diagnostic approaches, and clinical implications. **Methods:** Following PRISMA 2020 guidelines, five databases (MEDLINE, Web of Science, CINAHL, Scopus, and EMBASE) were searched from inception to January 2024. Inclusion criteria encompassed case reports, case series, and observational studies on ULA in humans. Risk of bias was assessed using the Joanna Briggs Institute (JBI) checklist. Narrative synthesis was performed due to methodological heterogeneity. Results: Thirty-two studies (137 participants) were included. Right-sided ULA predominated (58%), with poorer prognoses due to mediastinal distortion. Cardiovascular anomalies (40%) were the most common comorbidity. Diagnostic modalities included chest radiography (85%), CT (70%), and bronchoscopy (25%). Schneider-Boyden scale was used to classify the included studies. Risk of bias assessment revealed 65% of studies as low risk, 28% as moderate, and 7% as high risk. **Conclusions:** ULA necessitates multidisciplinary management, particularly in cases with associated anomalies. Left-sided ULA correlates with better outcomes, emphasizing the role of early imaging. Limitations include reliance on case reports and inconsistent reporting of anatomical variants. Future research should adopt standardized classifications and longitudinal designs to improve evidence quality. Open science framework (OSF): 10.17605/OSF.IO/XVQSP.

## 1. Introduction

Lung agenesis is a rare developmental defect in which there is complete absence of one or both lungs, including the bronchi, bronchioles, vasculature, and respiratory parenchyma. Its incidence ranges from 1 to 2 per 10,000 live births [1]. It may be unilateral or bilateral. Patients with left-sided agenesis, which is more common, have a longer life expectancy than those with right-sided agenesis [2]. Bilateral pulmonary agenesis is extremely rare and is incompatible with extrauterine life [3].

The etiology of lung agenesis is thought to be an embryological defect of the lung or vascular tissues or from an in utero vascular event [4]. Lung agenesis represents a failure of the primitive foregut branches, or lung buds, to develop very early on in the first trimester, between weeks 3 and 7 of gestation [5].

Unilateral lung agenesis (ULA) can occur without the presence of tetralogy of Fallot or other congenital anomalies. The literature describes a wide variety of associated congenital anomalies seen with lung agenesis, with cardiovascular anomalies being the most common, occurring in 40% of patients [6]. This was followed by skeletal (30%), gastrointestinal (20%), tracheal stenosis (20%), and genitourinary (14%) anomalies [6].

Some patients with ULA can remain asymptomatic and be missed during the neonatal period, only being diagnosed later in life during workups for recurrent lung infections [7,8]. These patients may not have been associated with congenital anomalies like tetralogy of Fallot. Therefore, ULA can occur in the absence of other major congenital defects, though associated anomalies are common and impact prognosis.

Thus, this review’s objective is to investigate the prevalence of ULA, considering its additional anatomic variants, and explore the correlation between ULA and their impact on symptomatology, clinical radiology, and surgical management.

## 2. Methods

### 2.1. Protocol

This systematic review and meta-analysis were conducted and reported in accordance with the Preferred Reporting Items for Systematic Reviews and Meta-Analyses (PRISMA) guidelines [9]. The systematic review was also registered with the open science framework (OSF) under registration number: 10.17605/OSF.IO/XVQSP. Additionally, we adhered to the protocol previously established by Orellana et al. (2024) [10].

### 2.2. Eligibility Criteria

Studies that reported ULA and its association with any clinical condition were considered eligible for inclusion if the following criteria were fulfilled: (1) Population: sample of dissections or images of the ULA. (2) Outcomes: ULA prevalence, additional variants, and their correlation with cardiopulmonary symptomatology. Additionally, anatomical variants were classified and described based on normal anatomy and classifications proposed in the literature. (3) Studies: this systematic review included research articles, research reports, or original research published in English or Spanish in peer-reviewed journals and indexed in some of the databases reviewed. Conversely, the exclusion criteria were as follows: (1) population: animal studies; and (2) gray literature such as reviews, letters to the editor or comments.

### 2.3. Electronic Search

We systematically searched in MEDLINE (via PubMed), Web of Science, the Cumulative Index to Nursing and Allied Health Literature (CINAHL), Scopus, and EMBASE from inception until January 2024.

The search strategy included a combination of the following terms: “Unilateral lung” (No Mesh), “Agenesia” (No Mesh), “Absence” (No Mesh), “Aplasia” (No Mesh) using the Boolean connectors “AND”, “OR” and “NOT”. The search strategies for each database are available in the Appendix A.

### 2.4. Study Selection

Four authors (MC-D; MB-S; JC-N; CC-O) independently screened the titles and abstracts of references retrieved from the searches. We obtained the full text for references that any author considered to be potentially relevant. We involved a fifth reviewer (MO-D) if consensus could not be reached.

### 2.5. Data Collection Process

Four authors (MB-S; MC-D; JC-N; CC-O) independently extracted data on the outcomes of each study. The following data were extracted from the original reports: (i) authors and year of publication, (ii) type of study, (iii) type of technique used to observe the ULA, (iv) sample characteristics (sample size, age, country, ethnicity, gender and laterality), (v) the ULA’s prevalence, and (vi) clinical/surgical implications.

### 2.6. Methodological Quality Assessment of the Included Studies

For the quality assessment of case study bias, four reviewers (MB-S; MC-D; JC-N; CC-O) independently performed data extraction and quality assessment. A fifth reviewer (MO-D) was involved if consensus could not be reached. To assess the risk of bias of case reports belonging to the category of descriptive studies, we used the Joanna Briggs Institute (JBI) critical appraisal checklist for case reports (last modified in 2017). Each article was evaluated using eight questions selecting the responses “yes,” “unclear,” “no,” or “not applicable.” Articles were evaluated using the following criteria: low risk of bias: more than 70% of the “yes” score, moderate risk of bias: 50% to 69% of the “yes” score, and high risk of bias: less than 49% “yes” score [11].

## 3. Results

### 3.1. Included Articles

The search resulted in a total of 6102 articles from different databases that met the criteria and search terms established by the research team. The filter was applied to the titles and/or abstracts of the articles in the consulted databases after duplicate records had been removed. Duplicates and other irrelevant records were eliminated using automated tools. In total, 3250 full-text articles were evaluated for eligibility for inclusion. Next, 1427 studies were excluded because their primary and secondary results did not match those of this review or because they did not meet the established criteria for good data extraction, resulting in 32 articles included for analysis with a total of 137 subjects/cadavers (Figure 1). Moreover, none of the included articles provided enough data to do a meta-analysis, due to the heterogeneity of the outcome measures.

### 3.2. Characteristics of the Included Studies and the Study Population

The systematic search yielded 32 eligible studies comprising 137 subjects with unilateral lung agenesis: 30 case reports and 2 case series [12,13,14,15,16,17,18,19,20,21,22,23,24,25,26,27,28,29,30,31,32,33,34,35,36,37,38,39,40,41,42]. Geographic distribution included 15 studies from the United States with 91 subjects (66.4%) [13,15,17,18,19,21,26,27,29,31,36,39,40,41,43], 4 from India with 7 subjects [12,22,32,37], 2 from Germany with 2 subjects [20,23], 2 from China with 20 subjects [24,38], 2 from Turkey with 2 subjects [30,33], and single studies from France with 4 subjects, Saudi Arabia with 1 subject, United Kingdom with 1 subject, Afghanistan with 6 subjects, Japan with 1 subject, Ukraine with 1 subject, and Denmark with 1 subject. Complete study-level characteristics are detailed in Appendix A.

Regarding laterality, right-sided agenesis was documented in 79 cases and left-sided in 58 cases, with laterality unspecified in the remaining cases due to insufficient reporting in the primary sources. Sex distribution showed 59 subjects with documented sex (37 female, 22 male), while 78 cases lacked sex specification in original reports.

Diagnostic approaches varied across studies, with chest radiography employed in the majority of cases [12,13,15,16,17,21,22,23,25,26,27,28,29,30,31,32,33,34,35,36,37,38,39,40,41,42], computed tomography widely utilized for anatomical characterization [12,13,15,16,17,21,22,23,25,26,27,28,29,30,31,32,33,34,35,36,37,38,39,40,41,42], and supplementary modalities including echocardiography, bronchoscopy, and MRI used for the comprehensive evaluation of associated anomalies [13,16,17,21,26,27,31,38,40].

Anatomical classification according to the Schneider-Boyden scale revealed Type I (complete absence) in 53 cases (38.7%), Type II (rudimentary bronchus) in 12 cases (8.8%), and Type III (hypoplasia) in 11 cases (8.0%). However, 61 cases (44.5%) could not be classified due to insufficient anatomical detail in the source publications, highlighting inconsistent reporting standards across the literature.

### 3.3. Methodological Quality Assessment of the Included Studies

For the analysis of bias risk for the articles included in the systematic review, out of the 32 articles, only the case series were evaluated for bias risk. Two authors independently applied this tool to each case report to arrive at an overall judgment with supporting justifications for each article, keeping away from the subjective judge [11]. Backer et al. [27] was judged as having a low risk of bias, indicating that the report met the majority of JBI criteria (≥70% “yes”) and therefore provided a clear description of patient demographics, clinical history, diagnostic methods, interventions and outcomes. In contrast, Zhang et al. [24] was assessed as having a moderate risk of bias (50–69% “yes”), with the table indicating incomplete reporting of post-intervention conditions and adverse or unanticipated events (items Q6 and Q7 marked not applicable), which reduced confidence in the completeness of outcome and safety data from that study. Overall, the assessment highlights the variability in bias risk among the included articles, with implications for the reliability of the findings presented in the systematic review (Appendix A).

### 3.4. Concomitant Anatomical Variants

In the reviewed studies there was an important number of concomitant anatomical variants, affecting multiple systems. These are tetralogy of Fallot (two subjects; 1.45%) [26,40]; intrauterine growth restriction (one subject, 0.73%) [26]; cerebral ventriculomegaly (one subject, 0.73%); [26]; dextrocardia (five subjects, 3.64%) [14,19,26,32,38]; renal unilateral agenesis (two subjects, 1.45%) [14,26]; heterotaxy syndrome (one subject, 0.73%) [26]; atrioventricular canal defect (two subjects, 1.45%) [26,40]; duodenal atresia (two subjects, 1.45%) [21,26]: Goldenhar syndrome (two subjects, 1.45%) [14,28]; right eye hypoplasia (one subject, 0.73) [28]; atrial septal defect (two subjects, 1.45%) [13,28]; patent ductus arterius (one subject, 0.73%) [28]; hiatal hernia (one subject, 0.73%) [29]; DiGeorge syndrome (one subject, 0.73%) [30]; persistent cloaca with an absent anal opening (one subject, 0.73%) [37]; small left-to-right shunt in the apical four-chamber (one subject, 0.73%) [38]; diaphragmatic hernia (three subjects, 2.19%) [17,18,40]; absence of the left side of the diaphragm (one subject, 0.73%) [42]; 13 ribs (one subject, 0.73%) [13]; club foot (two subjects, 1.45%) [14,18]; esophageal atresia (three subjects, 2.19%) [14,18,21]; hemivertebral fusion (three subjects, 2.19%) [L2–L3; T1–T2; T6–T7] [14,38]; cataract (one subject, 0.73%) [14]; tracheal stenosis (one subject, 0.73%) [16]; absent carina (two subjects, 1.45%) [16,23]; bilateral superior cava (one subject, 0.73%) [17]; biliary atresia (one subject, 0.73%) [17]; polyhydramnios (two subjects, 1.45%) [18,20]; tracheoesophageal fistula (one subject, 0.73%) [18]; short femur and humerus (one subject, 0.73%) [18]; and pelvic kidney (one subject, 0.73%) [18].

Some of the included studies documented patient survival to hospital discharge or through brief follow-up periods. In total, 11 individual case reports and small case series demonstrated successful short-term outcomes: Ahmed et al. [25] reported a 24-year-old woman alive at presentation; Gorla et al. [29] documented an infant who survived to discharge and remained well at six months; Colson & Mortelliti [36] described an 11-month-old who survived pneumonectomy and was alive at discharge; Uttarwar et al. [37] reported a neonate who remained asymptomatic at six-month follow-up; Russell et al. [39] documented survival to discharge with stability at two weeks; Biswas et al. [32] reported a 15-year-old who remained asymptomatic at one year; Nandalike et al. [13] described an 18-month-old who recovered completely after foreign body removal; He et al. [38] reported two pediatric cases that remained clinically stable; Tanrivermis Sayit & Elmi [33] documented a 53-year-old who was alive at short-term follow-up; Kumar et al. [12] reported a 40-year-old woman who improved and was discharged; and Upadhyay et al. [41] described a 28-year-old who survived with clinical improvement. These studies showed variability in follow-up duration, ranging from days to several months, and outcome measures were not standardized across reports. A smaller subset of studies provided documented evidence of sustained survival over multiple years. Krivchenya et al. [35] reported a child with right lung aplasia who was alive and well at a four-year follow-up after surgical intervention. Colson & Mortelliti [36] documented an infant who survived up to a five-year follow-up after pneumonectomy. Most notably, Kayemba-Kay’s et al. [14] reported four patients with ULA who were all alive and demonstrating normal growth and development at a mean eight-year follow-up period. Additional cases of long-term survival into adolescence and adulthood were reported by Nguyen [31] (14-year-old with mild exercise limitation), Tanrivermis Sayit & Elmi 2020 (53-year-old adult), and Thomsen [42] (36-year-old woman who had survived to adulthood). Several studies documented significant neonatal or early childhood mortality, particularly in cases involving right-sided agenesis or complex-associated anomalies. Weber et al. [26] reported three neonatal cases where two infants with right-sided agenesis and multiple anomalies died in the neonatal period, while one with left-sided agenesis survived but remained chronically oxygen-dependent. Backer et al. [27] documented mortality in 2 of 11 patients (18%) in their single lung group within a larger cohort of tracheal stenosis repairs. Zhang et al. [24] reported the largest prenatal series, documenting 18 fetuses with unilateral lung agenesis where 7 cases (38.9%) were confirmed by autopsy, indicating fetal or early neonatal death, while 11 cases (61.1%) survived to postnatal imaging. Multiple studies documented that corrective or palliative surgical procedures could achieve survival but were frequently associated with ongoing morbidity or chronic support requirements. Backer et al. [27] described tracheal reconstruction procedures in patients with single lung anomalies. Pierron et al. [15] reported two patients with right lung aplasia who underwent left pulmonary artery translocation, resulting in symptom improvement but requiring ongoing monitoring. Ito et al. [28] documented neonatal repair of total anomalous pulmonary venous connection in a patient with ULA who survived beyond one year. Labovsky et al. [40] described the management of a complex case involving tetralogy of Fallot, congenital diaphragmatic hernia, and complete left lung agenesis. Gopinathan et al. [16] reported a 13-year-old who required repeated tracheal interventions and ongoing nocturnal support. Several studies provided inadequate follow-up or outcome details to permit meaningful estimation of long-term survival or life expectancy. These included reports by Fitoz [30], Sadiqi & Hamidi [34], Thomas et al. [22] (which included one patient lost to follow-up), Bentsianov et al. [43], and Downard et al. [21]. These limitations significantly constrain the ability to establish comprehensive population-level survival estimates.

### 3.5. Clinical Considerations

The clinical considerations and associated findings reported in the included studies provide valuable insights into the correlations between unilateral lung agenesis and various clinical manifestations. Ahmed et al. [25] noted that patients with left-sided agenesis tended to have a better prognosis and longer life expectancy compared with those with right-sided agenesis. This is hypothesized to be related to the greater distortion in the tracheobronchial tree and mediastinal shifting that can occur with right pulmonary agenesis. In contrast, a more recent review by Nguyen et al. [31] suggested that the number of anatomical lobes and the ability of the remaining lung to adapt and meet the body’s oxygen demands may be a more important factor in determining the prognosis.

Congenital anomalies, including cardiovascular, skeletal, gastrointestinal, and genitourinary defects, were frequently reported in association with unilateral lung agenesis [13,17,21,26,28,29,30,31]. The presence of these associated malformations can significantly impact the clinical presentation and management of patients with ULA. Respiratory symptoms, such as recurrent infections, dyspnea, and wheezing, were commonly reported in the included case reports [13,16,31,32,41]. The severity of these symptoms may be influenced by the degree of lung hypoplasia, the presence of associated anomalies, and the compensatory mechanisms of the remaining lung.

In some cases, the ULA was an incidental finding, with patients remaining asymptomatic or presenting with only mild symptoms [33,37]. This highlights the importance of considering ULA in the differential diagnosis, even in the absence of overt respiratory symptoms. Overall, the clinical considerations and associated findings reported in the included studies emphasize the importance of a comprehensive evaluation and multidisciplinary management approach for patients with ULA, taking into account the potential impact of laterality, associated congenital anomalies, and respiratory function.

## 4. Discussion

This systematic review intended to determine the prevalence of the ULA, considering its additional anatomic variants, and explore the correlation between the ULA and their impact on symptomatology, clinical radiology, and surgical management. For this, we evaluated 32 articles comprising a total of 137 subjects to assess the clinical implications of ULA. Most of the included studies were case reports, with only two classified as case series, reflecting the descriptive nature of the available research on this rare condition.

Geographically, the included studies were conducted across a wide array of regions, with the United States contributing the largest number of subjects at 66.4% [26,27]. Other notable contributions came from countries such as India, China, Germany, Turkey, and several others, but the majority did not describe the ethnicities of their sample (cases), except for the studies by Ito et al. [28] (Asian) and the one by Bentsianov et al. [43] (Hispanic), which we recommend on future case reports or series since it might show a pattern. With respect to the gender distribution among the included studies, 62.5% were female and 15.63% were male, with 15.63% including both genders, and 6.25% of the studies did not specify gender, comprising 37 female and 22 male subjects among those with documented sex. This gender distribution underscores the need for further exploration into any potential sex-related differences in ULA presentation and outcomes.

Among the diagnostic techniques used by the included studies, the most frequently used were chest radiography and computed tomography (CT) scans for visualizing lung anatomy and identifying the absence of one lung [25,26]. In addition, several studies utilized echocardiography, bronchoscopy, and magnetic resonance imaging (MRI) to further detail the anatomical and functional characteristics of ULA and any associated anomalies [26,27]. These diverse methodologies facilitated a thorough understanding of ULA’s clinical and anatomical manifestations.

To understand the malformation of lungs, it is necessary to know something about their normal development. The first indication of lungs appears in embryos of about seventeen to twenty somites at an ovulation age of twenty-four days (Streeter’s Horizon XI). In the beginning, therefore, the lungs develop at a very high level, near the junction of occipital and cervical segments. Within two days (Horizon XII, twenty-five to twenty-seven days, embryos twenty-five to thirty somites), the primordium develops into a pear-shaped bulge of the ventral wall at the caudal end of a laryngotracheal groove. The factors that prevent the formation of the lung must become effective no later than this period (Horizon XII), early in the second month of menstrual age. Three different degrees of arrest of development may occur: (1) complete absence of one or both lungs (agenesis); (2) suppression of all but a rudimentary bronchus (aplasia); and (3) abortive growth (hypoplasia). Then, Boyden modifies these 3 degrees, including the consideration of the pulmonary vasculature in the scale, which Boyden classified as (1) complete absence of bronchus, lung parenchyma, and vasculature; (2) rudimentary bronchus is present, but complete absence of lung parenchyma and vasculature; and (3) hypoplastic lung, bronchus, or vasculature is present.

In the 32 studies included in this systematic review, we found that 53 patients had ULA type I (38.7%) (according to the Schneider Scale), 12 patients had ULA type II (8.76), 11 patients had ULA type III (8.03%), and in 61 patients, the study did not specify the type (44.52%).

### 4.1. Methodological Heterogeneity and Rigor of the Included Studies

The studies included in this systematic review display notable methodological heterogeneity, a common challenge in investigations of rare conditions such as ULA; the overwhelming predominance of single-case reports among the included papers underscores both the rarity of the condition and the consequent difficulty of conducting larger observational series or controlled studies, which in turn constrains the overall level of evidence [11]. Applying the Joanna Briggs Institute (JBI) critical appraisal checklist for case reports, Backer et al. [27] was judged to be at low risk of bias, indicating that this report provided sufficiently clear and comprehensive information on patient demographics, clinical history, diagnostic evaluation, interventions, and outcomes, thereby supporting greater confidence in the internal reporting quality of that case. In contrast, Zhang Y. et al. [24] was classified as having a moderate risk of bias, principally due to the incomplete reporting of key outcome elements (notably the post-intervention clinical condition and documentation of adverse or unanticipated events), which reducedd confidence in the completeness and interpretability of its outcome and safety data and warranted cautious integration of its findings into the synthesis. The remainder of the included literature consisted predominantly of individual case reports; given their single-patient descriptive design and the variable extent of reporting, these studies were not subjected to the same formal risk-of-bias appraisal applied to the above reports. In summary, the evidence base for ULA is dominated by single-case descriptions with substantial variability in methodological reporting and quality; although a minority of reports met the low-risk JBI criteria, overall heterogeneity and frequent omissions (clinical history, standardized anatomical classification, and outcome reporting) limited internal validity and precluded quantitative synthesis. Consequently, conclusions regarding prevalence, phenotype–outcome associations, and diagnostic or therapeutic effectiveness must be interpreted cautiously, and the degree of bias present likely attenuates the external generalizability of observed patterns [11].

### 4.2. ULA’s Variants Clinical Relevance

ULA presents significant clinical challenges due to its complex interplay with various congenital anomalies and its impact on respiratory function. This condition, characterized by the absence of one lung, often coexists with other congenital abnormalities, particularly cardiovascular defects, which occur in approximately 40% of cases [6]. For example, in the same paper, there was the presence of complete atrioventricular septal defect (AVSD) with moderate atrioventricular valve regurgitation and a hypoplastic left mural leaflet. Other abnormalities include skeletal (30%), gastrointestinal (20%), and genitourinary defects (14%).

These anomalies can complicate the clinical picture, influencing both the presentation and the management of ULA. The clinical manifestations of ULA vary widely, ranging from asymptomatic cases to severe respiratory distress. Patients with left-sided agenesis tend to have a better prognosis compared with those with right-sided agenesis. This disparity is likely due to the anatomical and physiological challenges associated with right-sided agenesis, which can cause significant distortion of the tracheobronchial tree and mediastinal structures [2,25]. The ability of the remaining lung to adapt and compensate for the missing lung tissue plays a crucial role in determining the clinical outcomes and life expectancy of these patients. In addition to respiratory symptoms such as recurrent infections, dyspnea, and wheezing, the presence of ULA can lead to significant morbidity due to its association with other congenital anomalies. These may include skeletal, gastrointestinal, and genitourinary defects, which further complicate clinical management and necessitate a multidisciplinary approach [26,31]. The included outcomes demonstrate considerable variability, with many patients surviving the immediate postnatal period and some achieving sustained multi-year survival. Among the included studies, we found two subjects (1.45%) [26,40] that presented ULA with a tetralogy of Fallot, which is the most common cyanotic congenital heart defect in childhood. Without surgical repair, survival is markedly limited, with 66% of patients reaching 1 year of age, 49% reaching 3 years, 24% reaching 10 years, and only 2% surviving beyond the fourth decade. Mortality by age 40 is approximately 95%, and survival past the sixth decade is exceedingly rare [44].

Surgical repair of tetralogy of Fallot has dramatically changed the natural course of the disease. In patients undergoing complete correction, overall survival rates reach 95% at 20 years, 92% at 30 years, and 88% at 35 years postoperatively. Despite this marked improvement compared with unoperated patients, long-term outcomes remain limited, as survival decreases to 77% at 40 years and 66% at 50 years after repair, with late mortality often related to right ventricular dysfunction, arrhythmia, and progressive heart failure [45].

Evidence regarding patients with tetralogy of Fallot concomitant with ULA remains scarce, and no robust data on long-term survival or life expectancy are currently available. Existing reports are limited to isolated case descriptions, underscoring the need for further studies to clarify prognosis in this rare association.

Moreover, intrauterine growth restriction (IUGR) compromises fetal oxygen and nutrient supply during critical phases of lung morphogenesis, producing persistent parenchymal, airway, and vascular abnormalities—fewer and enlarged alveoli, thickened septa and basement membranes, altered surfactant production, and impaired vascular growth—that reduce functional reserve and predispose to lifelong respiratory morbidity [46,47]. These perturbations of branching, alveolarization, and pulmonary vascularization provide a mechanistic framework linking IUGR to congenital respiratory malformations; by impairing early lung patterning and growth, IUGR may increase the risk of severe defects such as ULA, a focal failure of lung bud development that plausibly arises from the same disrupted intrauterine environment described above [46]. In this review, only one subject (0.73%) [26] was reported to have IUGR.

Pulmonary agenesis associated with dextrocardia is an exceptionally rare congenital anomaly. Prognosis is particularly poor in right-sided agenesis due to severe mediastinal shift and increased susceptibility to respiratory infections, with nearly half of patients dying before age five and overall mortality estimated at around 30% [25,26,32]. In this context, dextrocardia is usually secondary to lung absence rather than an isolated cardiac defect, and its coexistence with agenesis indicates greater structural severity and clinical complexity. A study by Krisnanda et al. [46] describes the case of a 26-year-old woman with right pulmonary agenesis and dextrocardia who survived into adulthood, demonstrating that, in the absence of recurrent infections and with adequate hemodynamic compensation, survival is possible. These exceptional cases reflect the great phenotypic heterogeneity of this condition and underscore the importance of early diagnosis and multidisciplinary follow-up. In summary, the available evidence suggests that life expectancy in patients with pulmonary agenesis and dextrocardia is limited during childhood, particularly when the right lung is affected. However, the presence of documented adult cases indicates that long-term survival is possible, although rare, and depends on the absence of associated malformations and on adequate clinical management.

Cerebral ventriculomegaly is defined as an enlargement of the atrium of the lateral cerebral ventricles and occurs in up to 2 per 1000 births. In a study of 213 patients, the survival rate was 85.6% to discharge for all infants [48]. There are no more studies of a concomitant ULA and cerebral ventriculomegaly, but in this study, one patient had a concomitant ULA and cerebral ventriculomegaly, who died at 38 days of life [26].

The unilateral renal agenesis was a concomitant anatomical variant in two subjects; this is a congenital defect that leads to being born with only one kidney and in normal situations, there is no clinical manifestation of this. In a study of 157 patients, 114 (73%) were alive, 43 patients (27%) died, and six deaths were caused by renal failure. The study concluded that survival from the day of unilateral renal agenesis diagnosis was similar to the expected survival adjusted for age, sex, and year of birth [49]. There are no more cases in the literature of a concomitant ULA and unilateral renal agenesis, but in the studies that we included, there are two cases of unilateral renal agenesis, where there was a 50% rate of survival: a patient died at 38 days of life, and the other patient remained alive in the 8 years of follow-up of the study [14].

Heterotaxy, or isomerism, is an abnormality characterized by the atypical arrangement of thoraco-abdominal organs across the left–right body axis, with a prevalence of approximately 1 in 10,000 live births [50]. Treatment strategies for patients with isomerism depend on the severity of the associated cardiac and extracardiac lesions, with most cardiac operations being palliative due to the rarity of achieving normal anatomy and persistently high mortality rates [51].

Determining the incidence of isomerism is challenging, often being underestimated, especially for left isomerism, which can present with milder heart conditions that may not require surgical intervention. Notably, Asians exhibit a higher prevalence of heterotaxy (32%) compared with Western populations [51]. Prognostic data on the coexistence of ULA and heterotaxy is lacking; however, the prognosis for patients with complex cardiac lesions remains poor, with a 1-year mortality exceeding 85% for asplenia and over 50% for polysplenia [51]. Further research is urgently needed to enhance understanding of this rare condition.

Atrioventricular (AV) canal defect is a congenital malformation characterized by deficient atrioventricular septation and a common atrioventricular valve. Although no primary studies provide reliable estimates of life expectancy in unrepaired AV canal defects, surgical correction has dramatically improved prognosis. Following complete repair, estimated survival is approximately 85% at 10 years, 82% at 20 years, and 71% at 30 years [52].

More recent cohort data show that in-hospital mortality has fallen to below 1%, with a freedom from reoperation of around 92% at 5 years and approximately 87% at both 10- and 15-years post-surgery [53].

No data exist regarding life expectancy in patients with AV canal defect concomitant with ULA, but in our review, we identified a case with both anomalies who survived only 25 days [26].

Duodenal atresia is a congenital intestinal obstruction usually distal to the ampulla of Vater in the second portion of the duodenum. During the eighth to the tenth week of embryological development, errors of duodenal re-canalization are the main cause of duodenal atresia. In duodenal atresia, there is complete obstruction of the duodenal lumen. Duodenal atresia occurs in 1 in 5000 to 10,000 live births [54]. In a study, Duodenal atresia was identified in 169 patients; late complications occurred in 12% of patients with congenital duodenal anomalies, and the associated late mortality rate was 6%, which is low but not negligible. Follow-up of these patients into adulthood is recommended to identify and address these late occurrences [55]. There is no reported evidence on the prognosis between concomitant duodenal atresia and ULA, but in our revised bibliography, two patients had concomitant duodenal atresia and ULA: one patient was followed up until 9 months of age and is still alive and well [21], and the other died at 25 days of life [26].

Moreover, one patient had esophageal atresia [21], which comprises a spectrum of congenital anomalies characterized by a lack of continuity of the esophagus (atresia). Large studies have identified the prevalence of esophageal atresia to be 2.3–2.4 cases per 10,000 births. In the last decade, survival has increased to 91–98%. Esophageal atresia has a high incidence of associated anomalies, ranging from 31% to 59%. Mortality is primarily related to associated anomalies and genetic syndromes, reducing survival to as low as 51%, and resulting in increased overall morbidity in the short- and long-term [56]. There is no literature on the prognosis between esophageal atresia and ULA, but in the three cases that we found, two patients were delivered at term and survived [14,18], and the last one was followed up until 9 months of age and is still alive [21].

Goldenhar syndrome, also known as oculo-auriculo-vertebral spectrum (OAVS), is a congenital malformation affecting approximately 1 in 3000 to 5000 newborns, with a male-to-female ratio of 3:2. Its classic features include mandibular hypoplasia with facial asymmetry, oculo-auricular malformations, and vertebral anomalies [57].

Literature including ULA and Goldenhar syndrome were not found, mainly because it is an extremely rare condition.

However, a case report describing a patient with Goldenhar syndrome, ULA, and total anomalous pulmonary venous connection (TAPVC) was found.

In this case report, the subject was diagnosed prenatally at 25 weeks and the neonate underwent corrective surgery on day 3 of life. At the one-year follow-up, echocardiography and computed tomography demonstrated unobstructed pulmonary venous return and preserved biventricular function.

Thus, this patient represents the first documented survivor beyond one year with the triad of Goldenhar syndrome, ULA, and TAPVC after neonatal repair.

Long-term prognosis will hinge on ongoing cardiopulmonary monitoring and any additional interventions, but the successful initial outcome suggests a favorable future trajectory [28].

Right eye hypoplasia, which is most often described clinically as microphthalmia or anophthalmia, and ULA are rare congenital anomalies. ULA carries a poorer prognosis when affecting the right lung, with overall survival rates of about 60% at two years, especially in the presence of associated malformations [6,26]. To date, no published evidence reports right eye hypoplasia and ULA occurring in the same patient. However, several cases document microphthalmia with ULA, supporting an indirect relationship [58,59]. The most plausible developmental link is disruption of the retinoic acid signaling pathway, where mutations in STRA6 and RARB give rise to the PDAC/Matthew–Wood syndrome, characterized by microphthalmia, pulmonary hypoplasia/agenesis, and diaphragmatic defects [60,61].

On the other hand, we found two subjects that presented ULA with an atrial septal defect (1.45%) [13,28]. An atrial septal defect (ASD) is one of the most prevalent congenital heart diseases (CHDs), ranking second after ventricular septal defects (VSDs) [62]. The condition involves a defect in the atrial septum that can lead to significant hemodynamic consequences if left untreated. Despite advancements in diagnostic and therapeutic strategies, the management of an ASD remains challenging due to variations in clinical practice guidelines (CPGs). A recent scoping review highlighted that many existing CPGs for ASDs lack systematic reviews and fail to meet acceptable quality standards, emphasizing the urgent need for evidence-based recommendations [62]. This underscores the importance of improving CPGs to enhance patient outcomes and standardize care approaches in ASD management [62]. The case report notes an accompanying atrial septal defect (ASD) in the patient with right unilateral pulmonary aplasia; the paper reports no data or estimates on life expectancy related to ASDs when concomitant with unilateral lung aplasia [13]. In the case reported by Ito et al. [28], unilateral (right) pulmonary agenesis was accompanied by multiple cardiac and syndromic anomalies including supracardiac total anomalous pulmonary venous connection (TAPVC), a large ASD, a small VSD, patent ductus arteriosus (PDA), and clinical features of Goldenhar syndrome; surgical repair of the TAPVC and direct closure of the ASD at 3 days of age resulted in no pulmonary venous obstruction and survival beyond one year postoperatively, indicating that—within this single-case report—an ASD concomitant with unilateral lung agenesis did not preclude early surgical correction and short-term survival, but the report provides no long-term life-expectancy data beyond the one-year follow-up [28].

Tracheoesophageal fistula (TEF) is a foregut malformation characterized by an abnormal communication between the trachea and the esophagus. Surgical repair of this anomaly has significantly improved long-term outcomes. In a large, multi-institutional cohort study of 6466 live-born infants with TEF, survival rates were approximately 89% at one month, 84% at one year, and 83% at five years. Better outcomes were observed in cases without associated anomalies, with overall survival often exceeding 90% in specialized centers when no major anomalies were present [63].

Rare associations between TEF and ULA have been reported. One case of TEF and right ULA describes a patient who survived and was doing well at ten years of age following surgical management [64].

Another concomitant variant observed alongside ULA, was the hiatal hernia; in a cohort of 455 patients undergoing repair of giant hiatus hernia, postoperative survival was 90% at 5 years and 75% at 10 years, with a median survival of 192 months (≈16 years) [65]. In this systematic review this rare presentation was seen in one subject (0.73%) [29]; the case report does not state an explicit life expectancy for the infant, due to the nature of the study’s design, in which they describe clinical stability with discharge at 4 months and no pulmonary hypertension at 6 months; given isolated ULA with compensatory contralateral hyperplasia, long-term survival can be good if major associated anomalies are corrected, but prognosis depends on the severity of cardiac and other malformations. Similarly, we found three subjects (2.19%) [17,18,40] that presented ULA with a diaphragmatic hernia; this presentation has multiple factors that contribute to the combined presentation of diaphragmatic hernia and ULA, such as mechanical compression of the fetal lung by herniated abdominal contents that impairs lung growth, but intrinsic defects in pulmonary development also contribute, consistent with a “two-hit” hypothesis in which shared genetic and developmental pathways (e.g., FOG2, GATA4, COUP-TFII, and retinoic acid-related genes) disrupt both diaphragm and lung morphogenesis, thereby producing concurrent diaphragmatic and unilateral pulmonary anomalies [66]. Absence of the left side of the diaphragm was found in one subject (0.73%) [42], who was a 36-year-old woman who complained of fatigue and lassitude; alongside this anomaly, she had mediastinum and tracheal displacement to the affected side and dislocation of the abdominal organs upwards into the thorax, such as the gastric body and the splenic flexure of the colon, besides the ULA, which resembled a total atelectasis of the left lung, and an emphysematous right lung and a flattening of the right side of the thorax due to the lag of contralateral intrapleural presion but did not describe the subject’s follow-up [42]. We did not find more cases of hemidiaphragmatic absence.

Regarding the DiGeorge syndrome or 22q11.2 deletion syndrome, which is a multisystem disorder caused by a hemizygous microdeletion on chromosome 22q11.2 that typically arises de novo and is variably expressed; cardinal features include conotruncal congenital heart defects, palatal anomalies, thymic hypoplasia with immunodeficiency, hypocalcaemia from parathyroid dysfunction, neurodevelopmental delay, and elevated risk for psychiatric illness (notably schizophrenia) [67,68]. Clinical severity ranges from neonatal life-threatening cardiac and immunologic complications to milder adult phenotypes, since it has a lot of possible complications, and affected individuals require lifelong multidisciplinary surveillance (cardiology, immunology, endocrinology, and developmental/psychiatric care) because of increased perioperative morbidity and premature adult mortality [69]. We found one subject (0.73%) that presented this syndrome with the ULA, where imaging and clinical findings demonstrated an absence of the left pulmonary artery, a rudimentary blind-ending left main bronchus consistent with bronchial aplasia, and complete absence of identifiable thymic tissue; the patient also exhibited hypocalcemia, indicative of parathyroid dysfunction, T-cell lymphopenia, a right-sided aortic arch, and complex congenital cardiac defects (ventricular and atrial septal defects with a single atrioventricular valve) with mediastinal shift and contralateral lung overinflation, but due to the study’s design, no follow-up was made, so there was no data about its life expectancy [30] and we could not find the previous literature about the two anomalies.

The following concomitant variations also presented with ULA: right eye hypoplasia found in one subject (0.73) [28], club foot in two subjects (1.45%) [14,18], short femur and humerus in one subject (0.73%) [18], hemivertebral fusion in three subjects (2.19%; L2–L3, T1–T2, T6–T7) [14,38], cataract in one subject (0.73%) [14], and a supernumerary rib in one subject (0.73%) [13]; all of these concomitant anomalies did not present any mortality risk.

Persistent cloaca is a congenital malformation in which the rectum, vagina, and urethra converge into a common channel, accounting for approximately 10% of all anorectal malformations [70]. Over 80% of patients with persistent cloaca have associated urological anomalies, and those with a long cloaca channel often present with complex additional defects [70]. The prognosis for patients with persistent cloaca and its associated abnormalities, such as pulmonary agenesis, is guarded and requires close monitoring and management of the complex congenital defects, as the long-term outlook depends on the severity of the anomalies and the involvement of the normal organs [70]. This was observed in one patient (0.73%) [37] along with the uncommon association of ULA [70,71]; no data regarding its life expectancy was found in the available literature.

Absent carina was observed in two subjects (1.45%) [16,23]. In the case reported by Gopinathan and Guruswamy [16], the patient demonstrated multiple concomitant anomalies centered on the absent carina: critical proximal tracheal stenosis and ipsilateral bronchial agenesis, tracheomalacia with need for tracheostomy, two endotracheal stents complicated by recurrent granulation and restenosis requiring repeated balloon dilatation, and chronic respiratory dependence with nocturnal BIPAP and recurrent airway procedures. Cardiac malformations included dextrocardia and a common atrium with atrioventricular concordance and ipsilateral pulmonary artery hypoplasia/absence; craniofacial defects (micrognathia, cleft palate) produced feeding difficulties requiring gastrostomy. Similarly, the right-sided pulmonary aplasia case described by Abel [23] featured the absence of the right main bronchus and carina with compensatory hyperinflation and mediastinal shift of the contralateral lung, a left-curved trachea with concentric distal narrowing, recurrent bronchopulmonary infections, and failure to thrive; no intracardiac defect was identified in that patient. These reports exemplify how absent carina commonly coexists with complex airway, cardiothoracic, and systemic anomalies that substantially affect clinical course and prognosis [16,23].

Moreover, a notable case involving a four-month-old child with ULA and biliary atresia (one subject 0.73%) leading to liver failure requiring transplantation and a small patent ductus arteriosum and bilateral superior vena cava was noted. While the heart structure was otherwise normal, the bilateral SVC may require careful consideration in surgical planning due to its implications for venous return and possible alterations in hemodynamics. Despite the surgical risks, the child thrived post-transplant, indicating that isolated ULA may have a favorable prognosis, but associated anomalies can significantly impact life expectancy [17].

As the reader can infer by now, ULA is frequently accompanied by additional congenital anomalies that significantly influence prognosis; in three reported cases these included esophageal atresia with tracheoesophageal fistula (two cases), ipsilateral pulmonary artery hypoplasia/atresia, branchial cleft cyst, patent ductus arteriosus, renal ectopia (pelvic kidney), skeletal abnormalities (clubbed feet, supernumerary rib, shortened long bones), cleft palate and micrognathia, and partial VATER association, with polyhydramnios noted antenatally—outcomes depended primarily on contralateral lung function and the severity of the associated cardiopulmonary and systemic malformations [18]. Lastly, in the single case reported by Kalache et al. [20], prenatal right ULA was confirmed postnatally and was accompanied by a sinus venosus atrial septal defect; no other structural anomalies were identified, and chromosomal analysis showed a normal male karyotype (46,XY). The neonate initially did well but developed progressive respiratory insufficiency requiring reintubation; surgical measures including aortopexy and implantation of a right pleural tissue expander were performed to prevent recurrent cardiopulmonary crises related to mediastinal shift [20]. No other study was found to have this presentation; hence, data about patients’ life expectancy remains unclear.

However, neonatal mortality remains substantially elevated in complex cases, particularly those involving right-sided agenesis or significant associated cardiac, tracheal, or other congenital anomalies. The available evidence consistently indicates that prognosis depends heavily on three primary factors: laterality of the defect (with right-sided agenesis carrying worse outcomes than left-sided), the presence and severity of associated congenital anomalies, and access to specialized multidisciplinary surgical and neonatal intensive care. The literature reveals a significant gap in robust, population-level, long-term survival data, limiting the ability to provide definitive prognostic counseling.

The incidental discovery of ULA in asymptomatic individuals underscores the importance of considering this condition in differential diagnoses, even in the absence of overt symptoms [33,37]. The variability in clinical presentation highlights the need for a tailored approach to each patient, considering the laterality of the agenesis, the presence of associated anomalies, and the compensatory capacity of the remaining lung.

Synthesis of the available data indicates that anatomical variant and laterality are the principal determinants of clinical course: type I (complete absence) and right-sided agenesis are overrepresented among more severe presentations and are commonly accompanied by cardiovascular malformations that drive early morbidity, whereas left-sided and hypoplastic variants more often permit compensatory pulmonary adaptation and delayed or asymptomatic presentation. These findings support a prognostic framework in which anatomical classification and screening for concomitant cardiac anomalies should guide risk stratification and individualized management planning.

### 4.3. Limitations

This systematic review presents several methodological limitations that warrant consideration when interpreting the findings. The wide age range spanning from infants to adults represents an inherent limitation of case report methodology. We included this broad age spectrum because ULA has its embryological origin during early fetal development, specifically resulting from the failed development of the lung bud between the third and seventh weeks of gestation. Consequently, unless surgical intervention is undertaken, this congenital anomaly persists throughout an individual’s entire lifespan.

Moreover, for the search strategy, we relied on five databases for study identification, which may have resulted in the omission of studies published in specialized databases that could potentially be included in this systematic review. The predominance of case reports (93.7%) and case series limited our ability to conduct comparative effectiveness analyses between diagnostic approaches or treatment modalities. Many included studies provided inadequate summaries of clinical settings and participant demographics, with 44.5% of cases lacking proper anatomical classification according to the Schneider-Boyden scale.

Quality assessment revealed considerable methodological heterogeneity among included studies, with variability in methodological rigor (65% low risk, 28% moderate risk, and 7% high risk of bias) potentially influencing the reliability of the synthesized findings. The extreme rarity of ULA and predominance of case reports precluded formal meta-analytical approaches and statistical heterogeneity assessment. These methodological constraints underscore the challenges inherent in synthesizing evidence for ultra-rare congenital anomalies and highlight the critical need for standardized reporting protocols and multicenter collaborative studies to advance understanding of ULA’s clinical implications.

Overall, the clinical relevance of ULA lies in its diverse manifestations and significant impact on patient quality of life across all age groups. The condition presents unique challenges due to its embryological origin and lifelong persistence, with clinical presentations ranging from asymptomatic cases to severe respiratory distress [72]. ULA may remain asymptomatic until adulthood, with symptoms potentially including dyspnea, hemoptysis, or recurrent respiratory infections when they manifest [72,73]. Anatomical laterality significantly influences prognosis, with right-sided agenesis demonstrating poorer outcomes due to mediastinal distortion, while left-sided cases exhibit better compensatory mechanisms. The frequent association with cardiovascular anomalies (40% of cases) necessitates comprehensive screening and ongoing surveillance.

Effective management requires early recognition through appropriate imaging, with radiological investigations like chest radiographs being useful in diagnosis, though computed tomography provides more definitive assessment [72]. A coordinated multidisciplinary approach involving pulmonologists, cardiologists, and surgeons is essential to address the multifaceted challenges presented by this condition. Long-term follow-up is crucial to monitor compensatory lung growth and manage complications such as recurrent lower respiratory tract infections and potential pulmonary hypertension. The rarity of ULA underscores the importance of centralized care in specialized centers, while standardized management protocols and patient registries could significantly improve outcomes and facilitate future research for this challenging congenital malformation.

Overall, the review’s interpretability is constrained by the rarity of ULA, reliance on retrospective case reports/series, incomplete anatomical and demographic reporting in nearly half of the cases, and potential publication and database-selection biases; these limitations prevent meta-analysis, hinder assessment of diagnostic and therapeutic comparative effectiveness, and underscore the urgent need for standardized reporting templates and multicenter registries to produce more robust, generalizable evidence

## 5. Conclusions

ULA is a rare congenital anomaly characterized by the complete absence of one lung, often accompanied by cardiovascular, skeletal, or gastrointestinal malformations. This systematic review synthesizes evidence from 32 studies (137 subjects) to evaluate ULA’s prevalence, anatomical variants, and clinical implications. Right-sided ULA predominated (58%), correlating with poorer prognoses due to mediastinal distortion and tracheobronchial compromise. The Schneider-Boyden classification revealed three anatomical variants: type I (complete absence, 38.7%), type II (rudimentary bronchus, 8.8%), and type III (hypoplasia, 8.0%), with 44.5% of cases unclassified. Left-sided ULA demonstrated better outcomes, likely due to reduced mediastinal shift and compensatory lung adaptation. However, 40% of the cases presented cardiovascular anomalies, necessitating multidisciplinary management. Methodological limitations, including a predominance of case reports (93.7%) and variability in anatomical reporting, underscore the need for standardized classification and longitudinal studies. Future research should prioritize large-scale cohorts to elucidate sex-related disparities (61% female prevalence) and optimize surgical or therapeutic interventions. This review highlights ULA’s clinical complexity and advocates for early diagnosis, comprehensive imaging, and collaborative care to mitigate morbidity in affected individuals.

## Figures and Tables

**Figure 1 diagnostics-15-02272-f001:**
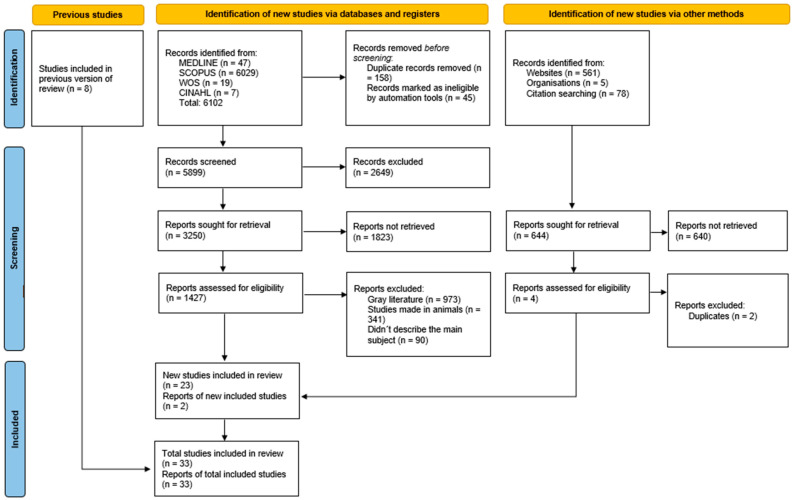
PRISMA 2020 search flow diagram for updated systematic reviews.

## Data Availability

Data can be found on the Appendix A.

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
