# Peer review of "Unilateral Lung Agenesis: A Systematic Review of Prevalence, Anatomical Variants, and Clinical Implications"

_diagnostics, 2025, doi:10.3390/diagnostics15172272_

Round 1
Reviewer 1 Report
Comments and Suggestions for Authors
This article systematically presents the epidemiological characteristics, anatomical classification, and diagnostic methods of unilateral lung agenesis (ULA), which has significant clinical implications. The writing of the article is well-structured and highly logical. Agree to publish, but suggest the following revisions:(1)Table 1 can be deleted, the relevant content can be summarized in words. (2)Table 2 and Figure 2 are also of no significant value. (3)The citation method for references in the article is inconsistent with that of this journal.
Author Response
Dear Reviewers,
We sincerely appreciate the time and effort you have dedicated to reviewing our manuscript. Your insightful comments and constructive feedback have been invaluable in refining our work and enhancing the clarity and impact of our findings. We are grateful for the opportunity to address your suggestions and provide further clarification where needed.
We have carefully considered each of your recommendations and have implemented the necessary revisions throughout the manuscript. To facilitate your review of the changes, all modifications have been highlighted in yellow in the revised version, allowing for easy identification of the amendments made in response to your valuable input.
Below, we respond to each of your comments point by point and outline the specific revisions made in the manuscript. We believe that these changes have significantly strengthened the quality and clarity of our work, and we hope that our responses adequately address your concerns.
We thank you once again for your thorough review and look forward to your consideration of the revised manuscript.
Sincerely,
Mathias Orellana.

Reviewer 2 Report
Comments and Suggestions for Authors
The authors performed a systematic review of very rare unilataral lung agenesis (ULA). The theme is very interesting, however, I felt they do not analyze well. They only gather and describe about published case reports.
First of all, it is very hard to read. They should use the number of reference instead of the author`s name and published year.
For explaining each parameters of the disease, they always describe the number of the reported cases and the authors of each reported data. But is it necessary? I want the authors to analyze and summarize the data.
Most of the gathered literature were case reports. I think it is not important to analyze the bias risk of each literature.
I want the authors to analyze and describe the feature of the ULA concesely.
Author Response

(The authors gave the same response as above.)

Reviewer 3 Report
Comments and Suggestions for Authors
This is a systemativ review that evaluted the prevalence, anatomic variants and clinical implicatons of unilateral lung agenesis. 32 studies with 137 participants were included in the analysis. The majority of the studes were case reports (30 out of 32 studies). The manuscript is focus primarly in
Some comments:
- The age range is wide (infants to adults). There are several limitations due to this: Symptoms, complications, and mortality risks differ drastically between neonates, children, and adults. Associated anomalies and survival rates may differ substantially between pediatric and adult cases. Please address this in the limitations sections.
- While ther anomalies are briefly mentioned, they are discussed without much detail. Please include the prevelence of other anomalies, such as tracheal stenosis, cardiovascular anomalies etc, in the tables and the text.
- The survival rates are not currently reported. Including survival data in the Results section would provide a more complete picture of prognosis.
Author Response

(The authors gave the same response as above.)

Round 2
Reviewer 2 Report
Comments and Suggestions for Authors
The authors revised the manuscript and it is much better than before now.
However, I think they can make it more concise.
The authors summarized the data better than before and used the reference number following to the former review, however, I think the authors still described too much about each case report in the latter paragraph of 3.4. There are repeated phrase like "Someone et al. describeded ~, someone et al documented ~ ." They described the contents of this paragraph in Discussion section better, so I think it can be reduecd more.
About gender distribution, they summarized well in the Line 149-50, so, I think the description from Line 287-290 is not necessary.
Line 317-331: They classified the cases into 3 types of Sneider`s classification, but I think the presentation of the number of each class is enough. The detail like "1 from Ahmed et al. , 3 from Weber et al. ~" is not necessary.
Author Response
Dear Reviewers,
We extend our sincere gratitude for the time and expertise you have generously invested in reviewing our manuscript. Your thoughtful observations and constructive critiques have proven instrumental in strengthening our work and substantially improving both the clarity and scientific rigor of our findings. We deeply appreciate the opportunity to address your valuable recommendations and provide additional clarification where warranted.
Each of your suggestions has been meticulously evaluated, and we have incorporated comprehensive revisions throughout the manuscript accordingly. To streamline your review process, all modifications implemented in response to your feedback have been clearly highlighted in yellow within the revised version, ensuring transparent identification of the amendments made.
In the following sections, we provide a detailed, point-by-point response to each of your comments, accompanied by explicit documentation of the corresponding manuscript revisions. We are confident that these substantial improvements have enhanced the overall quality, methodological soundness, and interpretability of our research, and we trust that our responses comprehensively address the concerns you have raised.
We remain deeply appreciative of your scholarly contribution to the peer-review process and respectfully request your consideration of our revised submission.
Sincerely,
Mathias Orellana
Reviewer 3 Report
Comments and Suggestions for Authors
Dear authors,
Thank you for your changes in the text accourding to the reviewers comments.
Author Response

(The authors gave the same response as above.)
